# Understanding the Metabolism and Dissipation Kinetics of Flutriafol in Vegetables under Laboratory and Greenhouse Scenarios

**DOI:** 10.3390/foods12010201

**Published:** 2023-01-02

**Authors:** María Elena Hergueta-Castillo, Rosalía López-Ruiz, Antonia Garrido Frenich, Roberto Romero-González

**Affiliations:** Analytical Chemistry Area, Department of Chemistry and Physics, Research Centre for Mediterranean Intensive Agrosystems and Agri-Food Biotechnology (CIAIMBITAL), Agrifood Campus of International Excellence (ceiA3), University of Almería, E-04120 Almería, Spain

**Keywords:** flutriafol, metabolites, dissipation, pesticide commercial product, laboratory, greenhouse

## Abstract

Flutriafol is a systemic triazole fungicide that is used to control diseases in various crops. A study was developed to evaluate the metabolism and dissipation of flutriafol in two different scenarios: laboratory and greenhouse conditions. Courgette and tomato samples treated with a commercial product (IMPACT^®^ EVO) at the manufacturer recommended dose were analyzed, and courgette samples were also treated at double dose. Ultra-high-performance liquid chromatography coupled with Q-Orbitrap mass spectrometry (UHPLC-Q-Orbitrap-MS), performing targeted and non-targeted approaches (suspect screening and unknown analysis), were used to analyze the samples. The dissipation of flutriafol was fitted to a biphasic kinetic model, with a persistence, expressed as half-life (*t*_1/2_), lower than 17 days. During suspect screening, three metabolites (triazole alanine, triazole lactic acid and triazole acetic acid) were tentatively identified. Unknown analysis led to the identification of four additional metabolites (C_16_H_14_F_2_N_4_, C_16_H_14_F_2_N_4_, C_19_H_17_F_2_N_5_O_2_ and C_22_H_23_F_2_N_3_O_6_). The results revealed that the proposed methodology is reliable for the determination of flutriafol and its metabolites in courgette and tomato, and seven metabolites could be detected at low concentration levels. The highest concentration of metabolites was found in the laboratory conditions at 34.5 µg/kg (triazole alanine). The toxicity of flutriafol metabolites was also evaluated, and some of them could be more toxic than the parent compound.

## 1. Introduction

Spain has over 50,365 ha of greenhouse used for horticultural production, and most of them are located in the Southeast region. The province of Almeria is known for having the largest land coverage of greenhouses in the world, reaching 48,389 ha in 2018 [1,2,3,4]. That year, the most widely grown crop was tomato (10,380 ha), which is the most cultivated product in Almeria, and courgette (or zucchini) was also one of the most cultivated crops with 7860 ha [1].

To control crops, plant protection products (PPPs) are widely used in agriculture to combat various diseases [5]. They consist of an active substance (pesticide) and other ingredients with different uses [6]. Flutriafol, [(R, S)-2, 4-difluoro-a-1H-1,2,4-triazol-1-ylmethyl)-benzhydryl alcohol, is a systemic triazole fungicide that acts by inhibiting the C-14α-demethylase enzyme involved in the biosynthesis of fungal sterols [7,8]. The use of this compound as fungicide has been approved since 2011 [7], but it was observed that it can cause lipid accumulation in human liver cells and rat liver through oxidative stress and apoptosis, and it may be a risk factor for liver diseases [9].

The persistence, expressed as half-life (*t*_1/2_), of flutriafol in soil is 1177.3 and 1587 days in field and laboratory conditions, respectively [10]. It has a high potential for mobility and may contaminate groundwater [11,12,13]. However, there is a lack of research on the persistence or dissipation of flutriafol in vegetables [11,14], using high-performance liquid chromatography coupled with tandem mass spectrometry (HPLC-MS/MS) [7]. To date, only two studies have focused on the dissipation of flutriafol in vegetables growing in greenhouses, and they only monitored the parent compound, whereas metabolites were not evaluated. Flutriafol was studied in tomato and cucumber samples using HPLC-MS/MS [10], and in grape samples using HPLC with ultraviolet (UV) detection [14]. Another study was carried out to determine the dissipation of flutriafol using HPLC-UV in soil [15].

The metabolism of flutriafol in primary crops has been investigated by monitoring four common metabolites, also known as triazole derivate metabolites (TDMs): triazole alanine (TA), triazole lactic acid (TLA), triazole acetic acid (TAA) and 1,2,4-triazole (1,2,4-T) [7]. The degree of formation of TDMs in plants and the environment depends on various factors, including the type of pesticide applied [16]. To protect public health, the European Union (EU) has set maximum residue limits (MRLs) for flutriafol at 0.8 mg/kg in tomatoes, and 0.15 mg/kg in courgettes [17]. These MRLs, like most pesticide limits, do not include information about metabolites but rather only the active substance [17]. Nevertheless, the European Food Safety Authority (EFSA) has set toxicological values (acute reference dose (ARfD)) for each TDM: 0.3 mg/kg bw for TA and TLA, 1.0 mg/kg bw for TAA and 0.1 mg/kg bw for 1,2,4 T, with the value for flutriafol being 0.05 mg/kg bw [18].

The only investigation that monitored TDMs from flutriafol and the parent compound in field conditions was performed on apple samples, detecting TA and TAA at trace levels (<0.001 mg eq/kg), while 1,2,4-T and TLA were not detected [7]. Even though TDMs have been described as metabolites of flutriafol at low concentration levels, the presence of other metabolites during the natural dissipation process of this fungicide in vegetables is still unclear. Therefore, it is necessary to perform an unknown analysis using a technique such as liquid chromatography-high resolution mass spectrometry (LC-HRMS) to obtain a full pathway of flutriafol.

Due to the lack of a comprehensive understanding of the behavior and dissipation kinetics of flutriafol and its degradation into its known and unknown metabolites, it was necessary to study flutriafol in two different scenarios. For that purpose, trials were carried out under laboratory (over 43 days) and greenhouse (over 55 days) conditions in courgette and tomato samples, respectively. The monitoring period in the greenhouse study was longer than in the laboratory because in the laboratory trials, courgette samples were treated with the PPP after they were collected, while in the greenhouse experiment, the tomato samples were treated with the PPP while still on the plant.

Courgette was selected for the laboratory trials for two reasons. First, to the best of our knowledge, the dissipation of flutriafol in this vegetable has not been reported yet, and, second, courgette has a different shelf-life than tomato. Tomato was chosen for the greenhouse scenario because there is a significant demand for this crop in Europe and it is the most important crop in Almeria, Spain [1]. Therefore, a rapid and sensitive analytical method for the determination of flutriafol in vegetables has been successfully established based on ultra-high-performance liquid chromatography (UHPLC) coupled with a Quadrupole-Orbitrap mass analyzer for targeted and non-targeted approaches (suspect screening and unknown analysis).

This study is the first one to evaluate the metabolism and dissipation of flutriafol in vegetables in two diverse scenarios, providing kinetics parameters of this fungicide and detecting known and unknown metabolites, taking advantage of the use of HRMS. The findings of this study are relevant for understanding the overall behavior of flutriafol in vegetables.

## 2. Materials and Methods

### 2.1. Material and Reagents

Flutriafol (CAS registry No. 76674-21-0, purity 99.54%) was supplied by Dr. Ehrenstorfer (Augsburg, Germany). The triazole acetic acid compound was purchased from Supelco (Buchs, Switzerland) (CAS registry No. 28711-29-7) with a purity of ≥98%. An individual stock standard solution (1000 mg/L) was prepared by dissolving 10 mg of the pure compound in methanol (10 mL). Intermediate solutions (10 mg/L and 1 mg/L) were prepared in methanol.

Methanol and acetonitrile, both LC-MS grade (99.9% of purity), were purchased from Honeywell Riedel-de-Haёn (Seelze, Germany). Formic acid (>98% of purity) was obtained from Panreac AppliChem (Barcelona, Spain) and water, LC-MS grade, was acquired from J.T. Baker (Deventer, The Netherlands). A mixture of acetic acid, caffeine, Met-Arg-Phe-Ala-acetate salt and Ultramark 1621 (ProteoMass LTQ/FT-hybrid electrospray ionization (ESI) positive) from Thermo Fisher Scientific (Waltham, MA, USA) were used for the accurate mass calibration of the Q-Orbitrap analyzer. The commercial pesticide product IMPACT^®^ EVO (12.5% flutriafol (*w*/*v*), SC, suspension concentrate) was purchased from FMC Agricultural Solutions, S.A.U. (Madrid, Spain), and it was manufactured by Cheminova A/S (Lemvig, Denmark).

### 2.2. UHPLC-Q-ORBITRAP-MS Equipment

Thermo Fisher Scientific Vanquish Flex Quaternary LC (Thermo Scientific Transcend™, Thermo Fisher Scientific, San Jose, CA, USA) was used for chromatographic analysis.

A Hypersil GOLD™ aQ column (100 mm × 2.1 mm; 1.9 µm particle size), supplied by Thermo Fisher Scientific, was employed. The flow rate was set at 0.2 mL/min, the injection volume was set at 10 μL and the column temperature was set at 30 °C. The mobile phase was composed of a water solution of 0.1% formic acid (eluent A) and methanol (eluent B) [19]. The eluent gradient was as follows: 0–1 min 95% A; from 1 to 4 min it decreased to 0% A and then the composition was kept constant for 6 min; finally, it returned to the initial conditions after 0.5 min and then remained constant for 3.5 min, for a total running time of 14 min [20].

The LC system was coupled with a hybrid mass spectrometer, Q-Exactive Orbitrap Thermo Fisher Scientific (Exactive™, Thermo Fisher Scientific, Bremen, Germany), using the electrospray interface (ESI) (HESI-II, Thermo Fisher Scientific, San Jose, CA, USA) in positive and negative modes. Different ESI parameters were set: spray voltage, 4 kV; sheath gas (N_2_, 95%), 35 (arbitrary units); auxiliary gas (N_2_, 95%), 10 (arbitrary units); S-lens RF level, 50 (arbitrary units); capillary temperature, 300 °C; and heater temperature, 305 °C. The mass spectra were acquired using four alternating acquisition functions: (1) full MS, ESI^+^, without fragmentation (the higher collisional dissociation (HCD) collision cell was switched off), mass resolving power = 70,000 full width at half maximum (FWHM), AGC target = 1 × 10^6^; (2) full MS, ESI^−^, without fragmentation (the higher collisional dissociation (HCD) collision cell was switched off), mass resolving power = 70,000 FWHM, AGC target = 1 × 10^6^; (3) data independent mass spectrometry fragmentation (DIA-MS/MS), ESI^+^ (HCD on, collision energy = 30 eV), mass resolving power = 35,000 FWHM, AGC target = 2 × 10^5^; (4) DIA-MS/MS, ESI^−^ (HCD on, collision energy = 30 eV), mass resolving power = 35,000 FWHM; AGC target = 2 × 10^5^. The mass range in the full-scan experiments was set to *m*/*z* 60–900 [19].

The chromatograms were acquired using the external calibration mode and then processed using Xcalibur™ version 4.3.73, with Quan Browser and Qual Browser, and Mass Frontier™ version 8.0 (Thermo Fisher Scientific, Les Ulis, France) to confirm the fragment ions by in silico approach. Furthermore, MassChemSite version 3.1 (Molecular Discovery Ltd., London, UK) was used to identify unknown metabolites.

### 2.3. Laboratory Trials

Fresh courgettes were purchased from a local market in Almeria (Spain). The study began immediately upon arrival at the laboratory. Dissipation experiments were performed from November to December 2021 under normal sunlight (at least 8 h sun exposure) in laboratory conditions.

To simulate normal crop conditions, courgettes were treated with the PPP at the recommended dose. Additionally, a higher application dose was evaluated to study the effect of the dose on the dissipation of the compound as well as to detect metabolites that may be present at low concentrations. Therefore, two concentration levels were tested: single dose (SD) and double dose (DD), which was two times the recommended dose. The single rate for the IMPACT^®^ EVO commercial product was 0.09%.

Samples were stored at room temperature (15 ± 5 °C) and were analyzed in triplicate with an approximate size of 500 g at 2 and 6 h, and at 1, 2, 5, 10, 15, 22, 29, 36 and 43 days. Some courgettes were established as blank, and they were not treated with the commercial product.

Furthermore, water loss was controlled by checking the weight of two courgettes (also considered as blank), and they were considered in the concentration calculation of the dissipation process.

### 2.4. Greenhouse Experiment

The greenhouse study was carried out in an agricultural greenhouse, using a hydroponic system located in the province of Almeria (Spain) over a period of 55 days. The field had not been treated with flutriafol or other fungicides with similar structures to triazole fungicides in the past five years. During the sampling period, from January to March of 2022, only this triazole fungicide was applied. The working field (156 plants distributed in 727.2 m^2^ in area) was divided into four blocks or plant lines (39 plants for each block). Three of them were sprayed with a 12.5% flutriafol emulsion at the recommended dose, and the other one, which was used as control, was not irrigated with the commercial product. Each block was divided into three plots, resulting in three replicates. Appendix A shows the characteristic parameters of greenhouse experiment.

Tomato samples were taken in triplicate with an approximate size of 500 g, and they were collected at 2 h and 1, 2, 3, 4, 7, 14, 24, 38 and 55 days after the treatment.

### 2.5. Sample Extraction

After collecting the samples, they were crushed and homogenized (including the peel). They were then analyzed using the solid-liquid extraction (SLE) procedure previously published [19]. This method involved adding 10 g of sample and 10 mL of acetonitrile to a 50 mL centrifuge tube. The mixture was then shaken for 1 min and centrifuged at 3700 rpm (3061 g) for 10 min. One mL of the supernatant, which had been previously filtered through a 0.2 μm filter, was collected and analyzed by UHPLC-Q-Orbitrap-MS.

### 2.6. Data Calculation and Analysis

The dissipation of flutriafol was evaluated by plotting residue concentration of this fungicide against time. The residual concentration (*C*(*t*)) and the half-life time (*t*_1/2_) were calculated using the biphasic kinetic model (Equations (1) and (2)). In these equations, *C*_0_ and *C*_1_ represent the concentration (µg/kg) in sample (courgette or tomato) at time t (day), *C*_0_ represents initial concentration of flutriafol (µg/kg) and *k*_1_ and *k*_2_ are the dissipation kinetic rate constants (days^−1^).
(1)$$C\left(t\right)=C_{0}e^{-k_{1}t}+C_{1}e^{-k_{2}t}$$
(2)$$t_{1/2}=\frac{\ln2}{k}$$

To estimate the toxicity of unknown metabolites, the Toxicity Estimation Software Tool (T.E.S.T), version 5.1.2, [21] was employed.

## 3. Results and Discussion

### 3.1. Dissipation Study of Flutriafol

The natural behavior and dissipation profile of flutriafol (through the treatment with IMPACT^®^ EVO commercial product) were evaluated. Flutriafol was monitored and its concentrations were measured at each sampling time and some differences were observed. It was noted that the general behavior of this fungicide during the study was an increase in concentration until it reached a maximum level and then decreased, as shown in Figure 1.

Different kinetic models (zero order, half order, first order and second order) were tested, and the biphasic model provided the best fit (R^2^ ≥ 0.986 in all cases). Figure 1a,b shows the dissipation of flutriafol at the laboratory trials for the two tested concentration levels (at SD and DD), while Figure 1c reports the results from the greenhouse experiment. These figures shown a similar behavior of flutriafol, where the dissipation of this fungicide was fitted to a biphasic kinetic model. The parameters of the kinetic model are listed in Table 1.

Although the degradation of pesticides generally follows first order kinetics [22], other studies have previously evaluated the dissipation of fungicides in tomato and courgette samples (in laboratory or greenhouse conditions) fitted to a biphasic kinetic model. For example, fenamidone and propamocarb were monitored in cherry tomato and courgette in greenhouse [23], and myclobutanil was studied in tomato samples under laboratory conditions [24], displaying the same degradation pattern.

#### 3.1.1. Dissipation under Laboratory Conditions

The lowest concentration of flutriafol in laboratory trials was observed six hours after treatment at both application doses. In this context, a concentration value of 407.9 µg/kg was observed in the case of the SD and 685.9 µg/kg when the DD was tested. In terms of initial concentration values, these revealed different information. As expected, the initial concentration of the SD (*C*_0_ = 385.9 µg/kg) was almost half the value of DD (*C*_0_ = 689.1 µg/kg). However, as shown in Table 1, similar degradation rates (*k*_1_ and *k*_2_ values) were reached: 0.05 and 0.06 days^−^^1^, respectively, for the SD, and 0.04 days and 0.07 days^−^^1^, respectively, for DD, with similar correlation coefficient values (i.e., R^2^ of (≥0.986)), indicating similar behavior in both scenarios.

The persistence or half-life (*t*_1/2_) was similar for the two doses. For the SD, the *t*_1/2_ was 14.8 days (*k*_1_) and 14.0 days (*k*_2_), whereas for the DD, the *t*_1/2_ was 16.3 days (*k*_1_) and 14.9 days (*k*_2_) (Table 1), indicating that, in the latter case, the dissipation process was slightly higher at the beginning. Thus, among the two concentrations, flutriafol was slightly persistent for the DD.

Based on these results, in which the dissipation process exhibited a similar trend, few differences were observed between both concentration levels under laboratory studies. Therefore, the applied dose of PPP did not affect the dissipation of flutriafol.

#### 3.1.2. Dissipation under Greenhouse Conditions

In relation to the greenhouse study (R^2^ = 0.966), the following values were obtained: the initial concentration (*C*_0_) of flutriafol was 315.6 µg/kg, the *t*_1/2_ was 8.9 days, and the degradation rates were 0.08 and 0.09 days^−^^1^, for *k*_1_ and *k*_2_, respectively, demonstrating few differences at the beginning and the end of the dissipation process (Table 1). Flutriafol was detected at concentrations between 55.0 (day 55) and 514.8 µg/kg (day 4). The residue level at the end of the monitoring period (55.0 µg/kg) was below the established MRL (800 µg/kg in tomato), indicating that the sample could be consumed.

It is worth noting that the dissipation process of flutriafol in vegetables did not conform to the few previous publications on greenhouse studies. In one of these studies, first-order kinetics (R^2^ ≥ 0.9708) was fitted to field tomato and cucumber samples treated with 25% flutriafol SC at DD [11]. The other study also followed first-order kinetics (R^2^ > 0.9385) in grape samples treated with 25% flutriafol SC [14]. When comparing the results of the kinetic parameters, the k constant found in these previous studies ranged between 0.07 and 0.08 day^−^^1^ in tomato and cucumber samples [11] and 0.09–0.10 day^−^^1^ in grape samples [14]. In the present study, this degradation constant was 0.08 day^−^^1^, which is a similar value to that obtained in the first study, while a slightly higher value was obtained in the second one, although the matrix was also different. Therefore, the degradation of flutriafol in grape was faster than in tomato samples. In terms of half-lives, 9.2–10.2 days [11] and 6.6–6.9 days [14] were achieved in each corresponding study. The first study showed the highest persistence in comparison to the other study and the present one. These findings might be explained by the fact that the same matrix (tomato) was evaluated in the first study, while grape samples were selected in the second one.

Additionally, the monitoring time of the current study in tomato samples was the longest carried out in field conditions, as flutriafol was monitored for 55 days compared to other studies in which the samples were only monitored for 28 days [11,14].

Finally, the observed differences may also be due to the agronomic conditions of each study, as climatic factors could affect the plant’s capacity of absorption, for instance [25].

### 3.2. Detection of Known Flutriafol Metabolites

The workflow used to identify flutriafol metabolites is shown in Figure 2, which was based on previous studies [26], and involves suspect screening and unknown analyses.

In addition to treating the samples with the recommended dose according to the manufacturer (SD), the laboratory samples were sprayed with a double dose (DD) to thoroughly monitor the metabolic pathway of flutriafol and evaluate any differences between the two concentration levels using a ‘suspect analysis mode’. The main reported metabolites of flutriafol were TDMs: TA, TLA, TAA and 1,2,4-T, and their chemical structures are shown in Figure 3. All of these TDMs were tentatively identified in the laboratory and greenhouse experiments except 1,2,4-T, which was not detected in either experiment. Characteristic parameters of TDMs are summarized in Table 2, showing that these compounds had low retention times (below 2 min) and suitable mass error (between −4.09 and 2.25 ppm).

For each metabolite, at least one fragment ion was monitored using Qual Browser and compared with the information provided by Mass Frontier™ software. The fragments were sorted according to the following criteria: the most abundant ion; retention time, which must be equal to the corresponding precursor ion; and mass error (lower than 5 ppm). Suitable mass errors for the fragment ions, with values between −1.96 and 2.28 ppm were achieved (Table 2). It should be noted that the ion at *m*/*z* 70.03997, which corresponds to the compound 1,2,4-T, was observed as the common fragment ion of two of these metabolites; TA and TAA, with −1.42 and 0.25 ppm of mass error, respectively, indicating adequate values. This ion is characteristic of triazole fungicides, and it is the common part of the structure of all of them [16].

The results obtained in the laboratory trials showed that the three TDMs (TAA, TA and TLA) were identified at different times during the study: TAA was the first to be detected, as it was found the first day after the application of the commercial product; TA was detected 2 days after treatment; and, finally, TLA was detected on the 15th day. There were significant differences between the results of the SD and DD experiments, as most of the compounds could only be detected when the DD of commercial product was applied, indicating a low concentration of the majority of the metabolites in the studied samples.

Appendix A shows the compound TA, which was detected at the two doses employed under laboratory conditions. Appendix A shows the extracted ion chromatograms (EICs) of TA after 48 h of treatment, illustrating that in both cases the corresponding peaks were found at 1.52–1.58 min for the SD and the DD, respectively. Additionally, the full-scan mass spectra of both concentration levels (Appendix A) were compared with the theoretical one (Appendix A), confirming the presence of this metabolite in the incurred samples.

In relation to the greenhouse experiment, TA was the first compound found, and it was detected 2 days after application, followed by TAA (day 7) and TLA, which was only identified 24 days after application.

Since a commercial analytical standard of TAA was available, the compound was successfully confirmed (Figure 4) by comparing retention times and MS spectra with those obtained by analytical standard, as well as by matching full-scan MS experimental with theoretical spectra. Figure 4a shows the EIC corresponding to TAA (after 7 days of application of treatment under greenhouse conditions) at 1.23 min, and Figure 4b shows the EIC of the analytical standard at 1.29 min, displaying similar retention times. The experimental (Figure 4c,d) and theoretical spectra (Figure 4e) show the similarities between both spectra.

Once the known metabolites were identified in the samples, their concentration was estimated. Therefore, a semi-quantitative approach was performed using the parent compound (flutriafol) for all metabolites, except for TAA, whose analytical standard (Figure 4b) was available.

In the laboratory trials, the concentration of the TDMs was found to be between 2.9 µg/kg for TAA (24 h after application) and 34.6 µg/kg for TA (at 48 h). TA, which showed the highest value of all TDMs, was detected at the beginning of the study (29.9 and 34.6 µg/kg for the SD and the DD, respectively). TAA was found one day after the treatment with the PPP at very low levels (2.0 µg/kg) and its concentration decreased over time. Regarding TLA, this metabolite showed concentration levels lower than the limit of quantification of flutriafol (LOQf), which was set at 2 µg/kg, both for the SD and the DD (on days 15 and 43 of the study). Therefore, it was noted that the behavior of TDMs decreased as the study progressed, and most of the compounds could only be identified (<LOQf) but not quantified.

The concentration of TDMs in greenhouse trials was 3.6 µg/kg for TAA on day 7. The other two metabolites (TA and TLA) could only be identified on one day of the study (days 24 and 38, respectively) and were not quantified due to the levels being below LOQf.

In conclusion, TDMs were found at trace levels in both the laboratory and in greenhouse scenarios, except for TAA, which was detected and quantified at concentrations between 2.0 and 3.6 µg/kg in both experiments. TA was also identified at levels > 30 µg/kg in laboratory conditions.

According to a study carried out by EFSA on TDMs in apple samples, in which 1,2,4-T and TLA were not identified and in which TA and TAA were detected at trace levels (<0.001 mg eq/kg) [7], considered with the results of this study, it is suggested that further investigation is needed to understand the plant metabolism of flutriafol. This study effectively addressed some of the limitations highlighted by EFSA by evaluating the pathway of flutriafol in two different matrices (courgette and tomato) under two different conditions and being able to identify and quantify TDMs.

In terms of TDMs toxicity, the most toxic compounds detected in this study, were TA and TLA, whose ARfD value = 0.3 mg/kg bw, although they are less toxic than the parent compound: ARfD = 0.05 mg/kg bw [18].

### 3.3. Tentative Identification of Unknown Flutriafol Metabolites

An unknown screening was performed to identify unknown flutriafol metabolites that have not been reported by EFSA or in the literature. For that purpose, the raw files obtained for each experiment were processed using MassChemSite (Figure 2). This software provided information on the metabolic pathways of organic molecules produced in chemical reactions on HRMS. By using the exact mass of the parent compound, flutriafol, (*m*/*z* 302.10994) and its corresponding peak and retention time (7.42 min), the tool proposed different compounds with their own retention times, structures and exact masses. The following criteria was used to select unknown metabolites: appropriate chemical structures in comparison with flutriafol. As a result, a total of four compounds were identified as new metabolites of flutriafol: C_16_H_11_F_2_N_3_ (flutriafol 1, referred to as MF1), C_16_H_14_F_2_N_4_ (flutriafol 2, referred to as MF2), C_19_H_17_F_2_N_5_O_2_ (flutriafol 3, referred to as MF3) and C_22_H_23_F_2_N_3_O_6_ (flutriafol 4, referred to as MF4), as shown in Figure 3. Based on their corresponding structures and exact masses, MF1 metabolite may have been generated through a dehydration reaction of flutriafol, as a water molecule was lost when this metabolite was produced. MF2 may have been produced through the substitution of an OH group (-OH) with an amino group (-NH_2_). MF3 may have been formed through acetylation of a urea binding, and MF4 through glycosylation of the benzene ring, as shown in Figure 3.

The most relevant characteristic parameters of these metabolites are shown in Table 3, in which two fragment ions were acquired for each metabolite in positive mode, with mass errors ranging from −2.86 to 4.45 ppm. MF2 was the first metabolite to elute (1.57–1.65 min) and MF4 was the last one (6.74–6.97 min). In all cases, mass error was always less than 5 ppm (Table 3).

It should be highlighted that MF1, MF3 and MF4 were detected under laboratory conditions, and that MF2, MF3 and MF4 were detected in the greenhouse experiment. This may be due to the different matrix used in each case or the differences in agronomic factors. Therefore, MF3 and MF4, which correspond to C_19_H_17_F_2_N_5_O_2_ (*m*/*z* 386.14231) and C_22_H_23_F_2_N_3_O_6_ (*m*/*z* 464.16170), respectively, were identified in both experiments.

After these metabolites were tentatively identified, a semi-quantitative estimation was carried out, using the parent compound as the standard, to determine the concentration of these compounds.

In both laboratory trials (SD and DD), the unknown metabolites were identified at the end of the study, revealing that MF3 and MF4 were only detected at levels below LOQf. Regarding MF1, this compound increased its concentration from 2.7 (day 29) to 2.9 µg/kg (day 43), as the study progressed for the SD (Appendix A), and from 2.8 (day 22) to 5.6 µg/kg (day 43), for the DD, as is observed in Appendix A.

The detected metabolites in the greenhouse scenario are shown in Table 4. MF1 was identified at the beginning of the study, especially the first day after the application of PPP until the seventh day, at concentrations below LOQf, indicating a short half-life for this metabolite with very low concentration. MF3 was only found in two days in the middle of the experiment: at 4.1 µg/kg (day 14) and 3.9 µg/kg (day 24 day). The half-life of this metabolite, like MF1, was also short, but it was at higher values. Finally, MF4 was the most recurrent metabolite, and it was identified from the third day of the experiment until the end (2.4 µg/kg, day 38 and 4.4 µg/kg on the last day), with its concentration increasing as the study progressed (Table 4). Most of the compounds were detected at trace concentrations (<LOQf).

Finally, in order to evaluate the possible impact of flutriafol metabolism, the toxicity of the metabolites was evaluated. As the information for flutriafol and TDMs has been reported in literature, as described above, the toxicity of the unknown metabolites was estimated by predicting their lethal dose 50% (LD_50_) in oral rats. For MF1 and MF3, these values were 313.8 and 944.2 mg/kg, respectively. No values were set for MF2 and MF4. Based on these values and the LD_50_ of flutriafol being set at >1140 mg/kg [10], it seems that the unknown metabolites could be more toxic than the parent compound. However, further studies should be carried out to evaluate the health and environmental concerns of flutriafol and its metabolites, as toxicological data could not be predicted for all the new flutriafol metabolites.

## 4. Conclusions

For the first time, the dissipation of flutriafol was investigated in two vegetable samples under two different scenarios: laboratory (courgette samples) and greenhouse (tomato samples) conditions. The persistence (*t*_1/2_) of flutriafol was less than 17 days, and it was observed that the dissipation process was not significantly affected by the application doses used in this study.

The results confirmed that the proposed analytical methodology is effective and reliable for the detection of flutriafol and seven metabolites: three known compounds (TDMs) and four unknown compounds. All TDMs could be suitably identified in the samples except for 1,2,4-T. These metabolites were present at trace levels, with concentrations below LOQf in most cases. The unknown metabolites were described for the first time, with two of them (C_19_H_17_F_2_N_5_O_2_ and C_22_H_23_F_2_N_3_O_6_) being detected in both scenarios, highlighting the role of HRMS in these types of studies. Therefore, this study has successfully provided valuable information about the metabolism and dissipation of flutriafol in vegetables, describing an overall picture of this fungicide in two matrices under two different conditions, increasing the knowledge about triazole fungicides and related metabolites.

## Figures and Tables

**Figure 1 foods-12-00201-f001:**
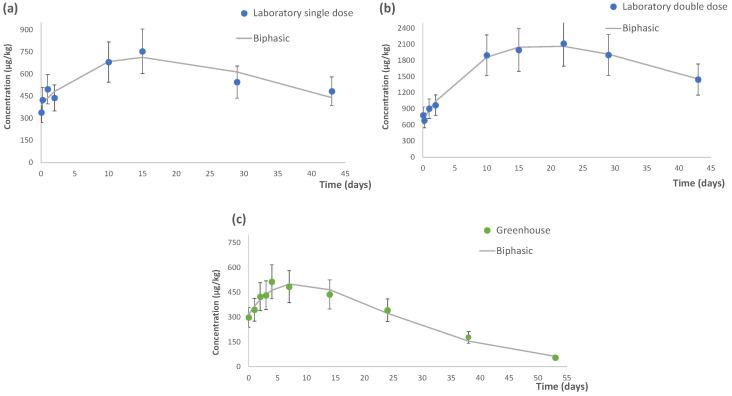
Biphasic kinetic model of flutriafol in laboratory trials at two concentration levels, (**a**) single dose (recommended dose by the manufacturer at 0.09 %) and (**b**) double dose (0.18 %), and (**c**) in greenhouse experiment (0.09 %) (*n* = 3).

**Figure 2 foods-12-00201-f002:**
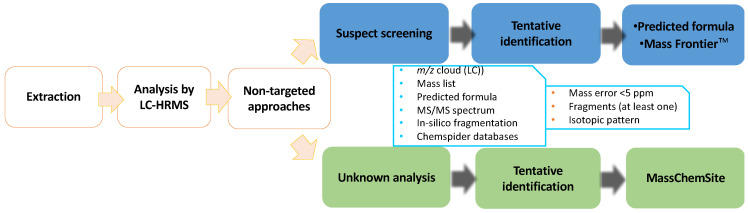
Workflow applied to the identification of metabolites by liquid chromatography-high resolution mass spectrometry (LC-HRMS).

**Figure 3 foods-12-00201-f003:**
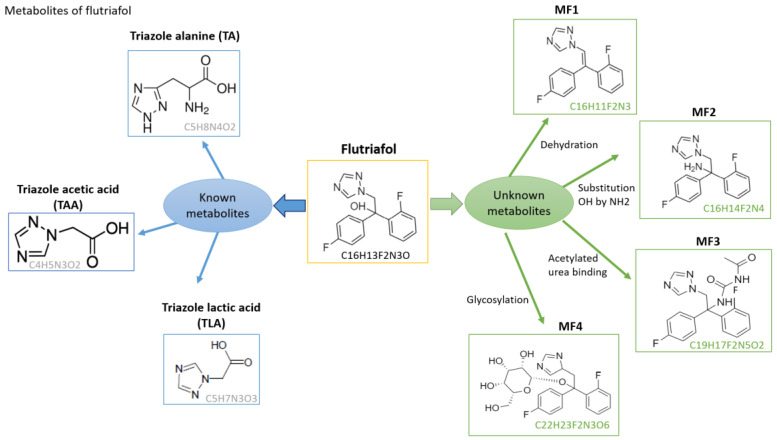
Metabolites of flutriafol identified by suspect screening and unknown analysis.

**Figure 4 foods-12-00201-f004:**
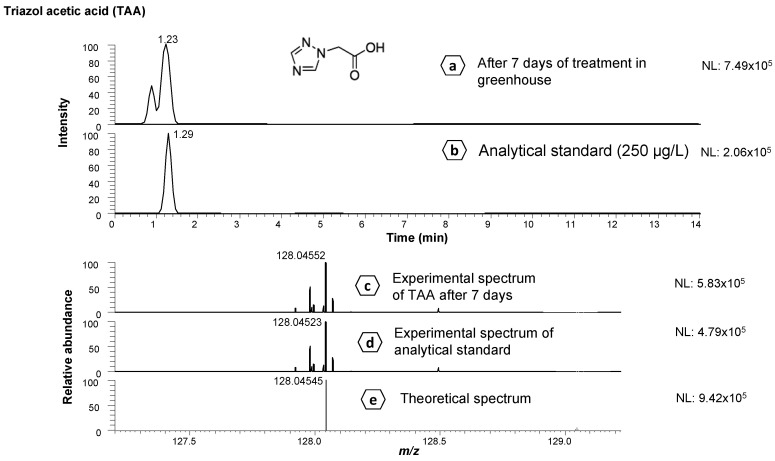
Extracted ion chromatograms of triazole acetic acid (TAA): (**a**) after 7 days of treatment in greenhouse; (**b**) analytical standard (at 250 µg/L); (**c**) full-scan MS experimental spectrum of TA after 7 days of treatment in greenhouse; (**d**) full-scan MS experimental spectrum of analytical standard; and (**e**) full-scan MS theoretical spectrum.

**Table 1 foods-12-00201-t001:** Biphasic kinetic model parameters of flutriafol dissipation in laboratory (courgette samples) and greenhouse trials (tomato samples) ^1^.

Kinetic Parameters	Laboratory Trials		Greenhouse Study (0.09%)
	Single Dose (0.09%)	Double Dose (0.18%)	
*C*_0_ (µg/kg)	385.9	689.1	315.6
*k*_1_ (days^−1^)	0.05	0.04	0.08
*k*_2_ (days^−1^)	0.06	0.07	0.09
*t*_1/2_(*k*_1_) (days)	14.8	16.3	8.9
*t*_1/2_ (*k*_2_) (days)	14.0	14.9	8.2
DT_90_ (days)	49.2	54.1	29.5
R^2^	0.986	0.995	0.966

^1^ Abbreviations: *C*_0_: initial concentration; *t*_1/2_: persistence or half-life; *k*_1_ and *k*_2_: dissipation kinetic rate constants; R^2^: correlation coefficient.

**Table 2 foods-12-00201-t002:** Characteristic parameters for tentatively identified metabolites of flutriafol by suspected analysis ^1^.

Compound	RTW (min)	Molecular Formula	Theoretical Mass (*m*/*z*)	Mass Error (ppm)	Fragment ions			Ionization Mode
					Molecular Formula	Theoretical Mass (*m*/*z*)	Mass Error (ppm)	
Triazole alanine (TA)	1.52–1.68	C_5_H_8_N_4_O_2_	157.07202	−0.18	C_2_H_4_N_3_	70.03997	−1.42	ESI (+)
					C_5_H_7_N_4_O	139.06144	−1.96	
Triazole acetic acid (TAA) CGA 142856	1.23–1.30	C_4_H_5_N_3_O_2_	128.04545	0.54	C_2_H_4_N_3_	70.03997	0.25	ESI (+)
Triazole lactic acid (TLA) CGA 205369	1.70–1.80	C_5_H_7_N_3_O_3_	158.05601	−4.09	C_4_H_7_N_2_O_3_	131.04512	2.28	ESI (+)
					C_5_H_5_N_3_O_2_	139.03763	1.41	

^1^ Abbreviations: ESI (+): electrospray interface in positive mode; RTW: retention time window.

**Table 3 foods-12-00201-t003:** Characteristic parameters for tentatively identified metabolites of flutriafol by unknown analysis.

Compound	RTW (min)	Molecular Formula	Theoretical Mass *(m*/*z)*	Mass Error (ppm)	Fragment Ions			Ionization Mode
					Molecular Formula	Theoretical Mass *(m*/*z)*	Mass Error (ppm)	
MF1	2.26–2.38	C_6_H_11_F_2_N_3_	284.09938	−1.12	C_9_H_11_N_2_F_3_	185.09114	4.37	ESI (+)
					C_4_H_10_O_5_N	152.05535	3.55	
MF2	1.57–1.65	C_16_H_14_F_2_N_4_	301.12593	−0.99	C_10_H_3_NF_2_	175.02281	4.45	ESI (+)
					C_14_H_11_NF_2_	231.08541	2.26	
MF3	3.37–3.54	C_19_H_17_F_2_N_5_O_2_	386.14231	1.45	C_8_H_10_N	120.08078	3.41	ESI (+)
					C_7_H_10_ON_4_	166.08491	2.82	
MF4	6.74–6.97	C_22_H_23_F_2_N_3_O_6_	464.16170	0.87	C_6_H_8_N	94.06513	3.40	ESI (+)
					C_12_H_20_NO	194.15394	−3.86	

**Table 4 foods-12-00201-t004:** Concentration (µg/kg) of unknown flutriafol metabolites under greenhouse conditions.

Metabolite	Sampling Period (Days)								
	1	2	3	4	7	14	24	38	55
MF2	<LOQ_f_	--	--	--	<LOQ_f_	--	--	--	--
MF3	--	--	--	--	--	4.1	3.9	--	--
MF4	--	--	<LOQ_f_	<LOQ_f_	<LOQ_f_	<LOQ_f_	<LOQ_f_	2.4	4.4

Abbreviations: MF2: metabolite 2; MF3: metabolite 3; MF4: metabolite 4; <LOQf: compound detected below limit of quantification of flutriafol (2 µg/kg) but not quantified; --: compound not detected.

## Data Availability

Data is contained within the article or supplementary material.

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
