# Peer review of "Understanding the Metabolism and Dissipation Kinetics of Flutriafol in Vegetables under Laboratory and Greenhouse Scenarios"

_foods, 2023, doi:10.3390/foods12010201_

Round 1

Reviewer 1 Report

In this work, authors detected Flutriafol and 7 metabolites though target and non-targeted approaches on the UHPLC-Q-Qrbitrap-MS. However, this experimental design is confusing. And it is noted that your manuscript needs careful editing by someone with expertise in technical English editing paying particular attention to English grammar, spelling, and sentence structure so that the goals and results of the study are clear to the reader.

Frist, why did the author choose courgette and tomatoes in the study, please explain clearly. And the author said that the two conditions were collected to understand the metabolism and dissipation of Flutriafol in fruits and vegetables, however, this study only choose courgette and tomato, and the laboratory experiment only selected the courgette and the greenhouse experiment only tomato, Can the author explain the relationship between them and the reasons for this choice?

Then, in the laboratory experiment, Is the courgette bought back from the market or is it grown in the laboratory? If the courgette was purchased at the market, how was it able to be stored at room temperature for 43 days? Won't they rot?

Finally, in the laboratory and greenhouse experiment, the test chemicals of Flutriafol were different, could the author please explain why? Differences in the metabolism and dissipation of the Flutriafol are indicated by whether the different test agents.

Please author carefully explains the above. The article may be considered for acceptance after the author has provided a reasonable explanation. And there are still some issues that need to be modified. A detailed list of comments and suggestions is reported below (following the order of appearance in the paper) to help the Authors during the revision stage.

- The metabolites of Flutriafol have been detected in this study, whether the authors considered going on to assess the toxicity of the four compounds (C16H14F2N4, C16H14F2N4, C19H17F2N5O2 and C22H23F2N3O6) and whether there was a continuing concern.

- The “DT50” should be changed to “T1/2”, which was often used to represent the half-life time.

- In the line 17-18, this sentence is not clearly expressed. In this, it may be better to describe T1/2 directly as less than 17 days.

- The toxicity and adverse impact of Flutriafol for animals and environmental and its metabolites should be provided and some relevant literature should be cited.

- 2.1. Equipment, material and reagents

Best organized into two paragraphs, one introduces the instrumentations, and another regent.

- 2.2. UHPLC-Q-ORBITRAP-MS analysis

The use of instrumentation of “Thermo Fisher Scientific Vanquish Flex Quaternary LC” should be introduces only in 2.1 Equipment, material and reagents.

“To estimate the toxicity of unknown metabolites, T.E.S.T (Toxicity Estimation Software Tool) software [20] was employed.” The software used for data processing should be placed in “2.6. Data calculation and analysis”.

- The line 130, the “adimensional” means? And N2 should be changed to “N2”.

- 2.3. Laboratory trials

In order to simulate natural crop conditions, courgettes were treated with the PPP at two concentration levels: single dose (SD) and double dose (DD), which was two times of the recommended dose, being the single rate for the IMPACT® EVO commercial product 0.09 L/hL.

Why different doses can simulate natural crop conditions? And the 0.09 L/hL should change to international units. Please note that all units mentioned in the text are to be in international units.

- The table 1. The captions should provide more detailed information, such as matrix.

- 3.1.1. Dissipation under laboratory conditions

“407.9 µg/kg, in case of SD and 685.9 µg/kg, at DD.” Please re-describe.

Please add the results of relevant literature for discussion for the dissipation Flutriafol.

- 3.3. Tentative identification of unknown flutriafol metabolites

If possible, please use a flow chart to represent the process of identifying an unknown compound. Please cite the literature correctly when applying to methods published by others

- Much of the conclusion is describing what was done in the paper and the conclusion is the answer, it needs to answer to be objective, that is, you need to write objectively the scientific findings of your paper.

Author Response

Reviewer 1

Comment: In this work, authors detected Flutriafol and 7 metabolites though target and non-targeted approaches on the UHPLC-Q-Qrbitrap-MS. However, this experimental design is confusing. And it is noted that your manuscript needs careful editing by someone with expertise in technical English editing paying particular attention to English grammar, spelling, and sentence structure so that the goals and results of the study are clear to the reader.

Response: English version of the manuscript was carefully revised.

Comment: First, why did the author choose courgette and tomatoes in the study, please explain clearly. And the author said that the two conditions were collected to understand the metabolism and dissipation of Flutriafol in fruits and vegetables, however, this study only choose courgette and tomato, and the laboratory experiment only selected the courgette and the greenhouse experiment only tomato, Can the author explain the relationship between them and the reasons for this choice?

Response: We do appreciate your comment and a clarification was included in the revised version of the manuscript (Introduction Section).

Courgette and tomato are among the most cultivated horticultural products in Spain. Therefore, they were selected as representative matrices in which flutriafol was applied. Thus, courgette was chosen to monitor the dissipation of flutriafol in the laboratory scenario, considering that it was not evaluated previously. On the other hand, tomato was selected for the greenhouse experiment due to there is a significant European demand for this crop and because it is the most cultivated product in Almería (Spain).

Therefore, this study provides new knowledge related to flutriafol dissipation, offering relevant information of this fungicide in two diverse scenarios, and evaluating two different matrices. 

Comment: Then, in the laboratory experiment, Is the courgette bought back from the market or is it grown in the laboratory? If the courgette was purchased at the market, how was it able to be stored at room temperature for 43 days? Won't they rot?

Response: More data were included in Section 2.3. Courgettes were purchased at the market and they were stored in the lab at 15 ±5ºC (November-December). Only loss of water was observed during that period and this fact was taking into account for the estimation of the concentration, as it was indicated in the manuscript.

Comment: Finally, in the laboratory and greenhouse experiment, the test chemicals of Flutriafol were different, could the author please explain why? Differences in the metabolism and dissipation of the Flutriafol are indicated by whether the different test agents.

Response: The same procedure was carried out in both scenarios (laboratory and greenhouse), because both targeted and non-targeted approaches were applied. Seven metabolites were detected in the study, and 5 out of 7 were found in both scenarios: 3 triazole derivate metabolites (TDMs) and 2 unknown metabolites, i.e., MF3 and MF4. Regarding the metabolites MF1 and MF2, MF1 was only detected at lab scenario and MF2 in the greenhouse one. The differences could be caused by the different matrix used in each case or by the differences between the agronomy factors as it was indicated in the revised version of the manuscript.

Comment: The metabolites of Flutriafol have been detected in this study, whether the authors considered going on to assess the toxicity of the four compounds (C16H14F2N4, C16H14F2N4, C19H17F2N5O2 and C22H23F2N3O6) and whether there was a continuing concern.

Response: Several flutriafol metabolites (referred as unknown flutriafol metabolites) could be detected in the present study, and due to it was the first time that they were identified, their toxicity values were predicted using the software tool provided by EFSA, as it was indicated in the manuscript. Despite of knowing the toxicity of parent compound (flutriafol), the estimation of the toxicity of its unknown metabolites is important to check if they could be as toxic as flutriafol, and this may affect to both the food and environment safety.

Comment: The “DT50” should be changed to “T1/2”, which was often used to represent the half-life time.

Response: This was modified throughout the manuscript.

Comment: In the line 17-18, this sentence is not clearly expressed. In this, it may be better to describe T1/2 directly as less than 17 days.

Response: The sentence was modified according to reviewer’s indication.

Comment: The toxicity and adverse impact of Flutriafol for animals and environmental and its metabolites should be provided and some relevant literature should be cited.

Response: It was included in the Introduction Section and a new reference was included.

Comment: 2.1. Equipment, material and reagents. Best organized into two paragraphs, one introduces the instrumentations, and another regent.

Response: This section was renamed and separated in two sections: section 2.1, only including material and reagents, and section 2.2, named as “UHPLC-Q-ORBITRAP-MS equipment”.

Comment: 2.2. UHPLC-Q-ORBITRAP-MS analysis

The use of instrumentation of “Thermo Fisher Scientific Vanquish Flex Quaternary LC” should be introduces only in 2.1 Equipment, material and reagents.

Response: According to the previous comment, this section was renamed, and the equipment use for the LC-MS analysis was kept in this one, which was renamed.

Comment: “To estimate the toxicity of unknown metabolites, T.E.S.T (Toxicity Estimation Software Tool) software [20] was employed.” The software used for data processing should be placed in “2.6. Data calculation and analysis”.

Response: This sentence was placed in section 2.6 according to reviewer’s comment.

Comment: The line 130, the “adimensional” means? And N2 should be changed to “N2”.

Response: “Adimensional” was replaced by “arbitrary units”, which is commonly used in bibliography, and “N2” was used instead of “N2”.

Comment: 2.3. Laboratory trials. In order to simulate natural crop conditions, courgettes were treated with the PPP at two concentration levels: single dose (SD) and double dose (DD), which was two times of the recommended dose, being the single rate for the IMPACT® EVO commercial product 0.09 L/hL. Why different doses can simulate natural crop conditions? And the 0.09 L/hL should change to international units. Please note that all units mentioned in the text are to be in international units.

Response: We appreciate this comment and a clear explanation was included in section 2.3. Thus, a higher dose than the recommend one was selected to detect those transformation products that may be generated at lower concentrations as well as the effect of the dose on the dissipation of the compound. Moreover, the units of application doses have been modified according to the reviewer comment. Thus, “L/hL” was replaced by “%” which are easily recognized.

Comment: The table 1. The captions should provide more detailed information, such as matrix.

Response: The corresponding matrix was included in each case.

Comment: 3.1.1. Dissipation under laboratory conditions “407.9 µg/kg, in case of SD and 685.9 µg/kg, at DD.” Please re-describe. Please add the results of relevant literature for discussion for the dissipation Flutriafol.

Response: The sentence was modified according to the reviewer’s comment. The information related to the comparison of these results with those obtained in previous studies was also included.

Comment: 3.3. Tentative identification of unknown flutriafol metabolites. If possible, please use a flow chart to represent the process of identifying an unknown compound. Please cite the literature correctly when applying to methods published by others.

Response: A new figure, showing the workflow was included (new Figure 2). To clarify the whole procedure, it was added to Section 3.2, describing the main steps when suspect screening and unknown analysis were carried out.

Comment: Much of the conclusion is describing what was done in the paper and the conclusion is the answer, it needs to answer to be objective, that is, you need to write objectively the scientific findings of your paper.

Response: The conclusion section was rewritten.

Reviewer 2 Report

The submitted manuscript evaluates the metabolism and dissipation of flutriafol in fruits and vegetables in two different scenarios, providing kinetics parameters and detecting known and unknown metabolites by HRMS.

The manuscript is well organized and, in my opinion, only small changes described below are required to accept for publication.

Line 63: change “EFSA (European Food Safety Authority)” to “European Food Safety Authority (EFSA)”;

Line 64: change “ARfD (acute reference dose)” to “acute reference dose (ARfD)”;

Line 73: change “LC-HRMS (liquid chromatography-high resolution mass spectrometry)” to “liquid chromatography-high resolution mass spectrometry (LC-HRMS)”

Line 109: change to “electrospray ionization (ESI)”;

Line 128: change “an electrospray interface (ESI) (HESI-II, Thermo …” to “the electrospray interface HESI-II (Thermo …”;

Line 253: change R2 to R2;

Line 415: change “313.8 mg/kg and 944.2 mg/kg “ to “313.8 and 944.2 mg/kg“;

Line 423: change “persistence lower (DT50) than 17 days.” To “persistence (DT50) lower than 17 days.”.

Author Response

Reviewer 2

The submitted manuscript evaluates the metabolism and dissipation of flutriafol in fruits and vegetables in two different scenarios, providing kinetics parameters and detecting known and unknown metabolites by HRMS.

The manuscript is well organized and, in my opinion, only small changes described below are required to accept for publication.

Comment: Line 63: change “EFSA (European Food Safety Authority)” to “European Food Safety Authority (EFSA)”.

Response: This sentence was replaced according to the reviewer comment.

Comment: Line 64: change “ARfD (acute reference dose)” to “acute reference dose (ARfD)”.

Response: This sentence was corrected.

Comment: Line 73: change “LC-HRMS (liquid chromatography-high resolution mass spectrometry)” to “liquid chromatography-high resolution mass spectrometry (LC-HRMS)”

Response: This sentence was modified.

Comment: Line 109: change to “electrospray ionization (ESI)”;

Response: This sentence was corrected.

Comment: Line 128: change “an electrospray interface (ESI) (HESI-II, Thermo …” to “the electrospray interface HESI-II (Thermo …”;

Response: This was modified.

Comment: Line 253: change R2 to R2;

Response: This sentence was corrected.

Comment: Line 415: change “313.8 mg/kg and 944.2 mg/kg “ to “313.8 and 944.2 mg/kg“; 

Response: This sentence was modified.

Comment: Line 423: change “persistence lower (DT50) than 17 days.” To “persistence (DT50) lower than 17 days.”.

Response: This sentence was corrected.

Round 2

Reviewer 1 Report

 Extensive editing of English language and style required

Author Response

Comment: Extensive of English language and style required.

Response: English version of the manuscript was revised.